# Early opportunity signals of a tipping point in the UK's second-hand electric vehicle market

Chris A. Boulton[1*], Joshua E. Buxton[1], Timothy M. Lenton[1]

[1]Global Systems Institute, University of Exeter, Exeter, EX4 4QE, UK

*Correspondence to*: Chris A. Boulton (c.a.boulton@exeter.ac.uk)

**Abstract.** The use of early warning signals to detect the movement of natural systems towards tipping points is well established. Here, we explore whether the same indicators can provide early opportunity signals (EOS) of a tipping point in a social dataset - views of online electric vehicle (EV) adverts from a UK car selling website (2018-2023). The daily share of EV adverts views (versus non-EV adverts) is small but increasing overall and responds to specific external events, including

abrupt petrol/diesel price increases, by spiking upwards before returning to a quasi-equilibrium state. An increasing return time observed over time indicates a loss of resilience of the incumbent state dominated by internal combustion engine vehicle (ICEV) advert views. View share also exhibits increases in lag-1 autocorrelation and variance consistent with hypothesised movement towards a tipping point to an EV-dominated market. Segregating the viewing data by price range and year, we find a change in viewing habits from 2023. Trends in EOS from EV advert views in low-mid price ranges provide evidence that

these sectors of the market may have passed a tipping point, consistent with other evidence that second-hand EVs recently reached price parity with equivalent second-hand ICEV models of the same age. This pioneering analysis of how EOS applied to novel data can be used to predict the approach to a tipping point in a social system warrants further research to test the robustness and wider applicability of the method.

## 1 Introduction

Tipping points, where a small change in forcing can cause a large response in a system have been widely researched in a number of different fields, such as ecology (Scheffer et al., 2009) and climate dynamics (Lenton et al., 2008). More recently, there has been a focus on tipping points in society, particularly 'positive tipping points' that accelerate actions to combat climate change (Lenton, 2020; Otto et al., 2020).

One such positive tipping point to consider is a market switch from internal combustion engine vehicles (ICEVs) to electric vehicles (EVs) in light road transport. Historically, some behavioural and technology changes in the transport sector have been rapid, with the diffusion of innovation seemingly passing a tipping point (Rogers et al., 2014). For example, the transition from horse drawn carriages to ICEVs was relatively quick, happening over a decadal time span at the start of the 20th century

(Grübler et al., 1999; Nakicenovic, 1986), although there was a twenty year interval of change and competing technologies beforehand (Geels, 2005).

In the case of such 'S-curve' diffusion of innovations (Rogers et al., 2014), a tipping point can be defined in terms of a critical mass of adoption in a population, i.e. the point at which one more person adopting ultimately triggers the majority remaining to adopt in a self-propelling fashion. Critical mass tipping points can arise from several distinct reinforcing feedback mechanisms, as reviewed in e.g. Lenton et al. (2022b); Zeppini et al. (2014). For example, Rogers et al. (2014) describe the learning process whereby early adopters share knowledge with others in a population encouraging increased adoption (without any need for change in the qualities of the thing being adopted). Alternatively, 'increasing returns' describes how adoption can increase the pay off for the next adopter, thanks to learning curves for a technology (e.g. involving learning-by-doing, economies-of-scale) that improve its performance and reduce its price (Arthur, 1989).

There are several reasons to think that the car market could exhibit a tipping point from dominance by ICEVs to dominance by EVs. Empirically, such market tipping has already been observed over the last decade in Norway (Sharpe and Lenton, 2021), following a classic 'S-curve' pattern (Rogers et al., 2014; Dosi et al., 2019; Kucharavy and De Guio, 2011; Silverberg and Verspagen, 1994). Several other markets are also following an S-curve trend (e.g. Iceland, Sweden, the Netherlands, China) and globally EV sales are doubling every ~1.5 years, consistent with early S-curve growth. Theoretically, several reinforcing feedback mechanisms suggest EVs will come to dominate rather than coexist with ICEVs. Notably, learning-by-doing and economies-of-scale are making EV batteries both better and cheaper, the more that are produced. These increasing returns are making EVs more attractive and affordable than ICEVs in major markets, starting with China. Additionally, social contagion is happening in adoption of EVs, and technological reinforcement is happening between increasing EV adoption and increasing roll out of charging infrastructure.

Many interlinked factors could drive the market towards a tipping point to an EV dominated state, including declining cost of batteries, government policy incentives for EV adoption or forthcoming bans on ICEVs, car manufacturers switching strategy and technology investment, public and private investment in charging infrastructure, and increased public acceptance of EVs. Also, as the ICEV market and then the ICEV stock shrinks, fuel infrastructure will start to decline, making it less attractive and less convenient to own an ICEV in a self-propelling decline.

Early warning signals (EWS) of tipping points in natural systems refer to statistical indicators that a tipping point is approaching. They are based on the theory of critical slowing down (CSD): the phenomena that as the incumbent state (or attractor) of a system loses stability, it will respond more sluggishly to perturbations and take longer to return to equilibrium (Wissel, 1984). This is often referred to as loss of 'resilience' as dampening feedbacks that maintain the incumbent state get weaker. Here we assess CSD indicators of loss of resilience for the case of the incumbent state of a technology (here ICEVs)

dominating a market. The corresponding damping feedbacks are things that tend to maintain the status quo of ICEV market dominance, e.g. manufacturers and consumers resisting change. While EWS have been found in many natural systems approaching tipping points (Dakos et al., 2008; Dakos et al., 2023), it is less clear if EWS precede the movement towards societal tipping points, although some studies suggest they may exist in certain social systems (Brummitt et al., 2015; Neuman et al., 2011).

EWS generally assume a timescale separation; a long term, slow forcing towards tipping, and short term fluctuations which move the system around its equilibrium. If known perturbations occur, the rate of recovery of the system to its original state can be directly measured (Veraart et al., 2012). With a time series of the system, CSD can also be estimated by measuring the lag-1 autocorrelation (AR(1)) and variance over time on a moving window (Held and Kleinen, 2004) - with both statistical indicators predicted to increase if CSD is occurring (Ditlevsen and Johnsen, 2010). These indicators are detailed more in the methods section. They are referred to as early "warning" signals because they have principally been applied to undesirable tipping points in natural systems. Here, we look for them in the case of a more desirable tipping point in a social-technological system, as they are a way of measuring a loss of resilience in the ICEV dominated market. Hence we refer to them as early "opportunity" signals (EOS) – because if they exist, policymakers, firms, investors and other actors could conceivably use them to steer further interventions to deliberately trigger tipping (Lenton, 2020).

Here we explore the possibility of predicting the movement towards tipping in the UK's second hand car market from an incumbent state of high interest in ICEVs to one of high interest in EVs, using EV online advert views on Auto Trader UK's website as a social sensing tool to gauge public interest. The EV market in the UK has grown over the last few years, with new EV sales making up 0.7% of all vehicle sales in 2019, up to 16.5% in 2023 (Auto Trader Uk, 2023). Specific to our analysis, interest in EVs is also growing; 0.9% of all advert enquiries on Auto Trader were for EVs in 2019, however this had increased up to 7% by 2023, the share doubling from 3.5% in 2022 (Auto Trader Uk, 2023). In the next section we discuss our data sources and EOS methods. Then we present our results. Finally, we discuss their significance.

## 2 Data and Methods

### 2.1 Auto Trader Adverts and Views

Our data is taken from Auto Trader UK, a car selling website that allows both private sellers and car dealerships to advertise cars. We have used Auto Trader data as it is the UK's largest automotive marketplace, with over 75% of all minutes spent on automotive classified sites.

We focus on Auto Trader EV (full battery electric) adverts which have their engagement tracked from when the seller creates the advert. We have data on the daily number of views each advert has had, as well as its advertised price, from the start of 2018 up to July 2023. The dataset is split to consider both new and used cars, although only ~6% of adverts available on a given day are for new cars, thus our results here mainly apply to the second-hand EV market. Across the time period, 1.3% of used car advert views are on EV adverts, compared to 9.6% of the new car advert views. We consider the marketplace in two ways. First, to understand the marketplace as a whole, we explore the daily share of views that were for EV compared to non-EV advert views. We then view different niches in the market by separating the data into different price bands, to determine if there are changes in attention in EVs of certain price ranges.

## 2.2 Early Opportunity Signals

We begin by measuring the return time from a perturbation within a time series. To measure this, we look for the amount of time it takes for the time series to return 75% of the way to the minimum of the 10 days prior to perturbation. We originally considered calculating the return rate of the views as in Lenton et al. (2022a). However, particularly for the first spikes, the fit of an exponential decay model was poor due to a fast post-spike decline. We found return time to be a better measure for this specific data.

Then, we explore two EOS indicators that are calculated on a moving window across a time series: AR(1) and variance. As these assume stationarity, the time series are first detrended using a kernel smoother (using a bandwidth of 50 unless otherwise stated), and then the indicators are calculated on the residuals, using a moving window (2 years as standard) which slides across the residual time series, creating a time series of the indicator itself. From this, the tendency in each indicator can be measured, for which we use Kendall's $\tau$ correlation coefficient which equals 1 if the time series is always increasing, -1 if the time series is always decreasing, and 0 if there is no overall trend. Mathematically, an increase in these indicators is predicted if CSD is occurring (e.g., Ditlevsen and Johnsen (2010)). They can increase for other reasons so it is important to have independent evidence of the potential for a tipping point, which we take from the observed tipping of the car sales market in Norway towards EVs.

We calculate the significance of an obtained $\tau$ correlation coefficient through the use of null models (Dakos et al., 2008). We randomly reorder the time series of residuals and calculate our EOS on this 10000 times, determining a p-value by calculating the proportion of these that are above our AR(1) or variance $\tau$ from the original time series. These null models are designed to destroy the memory in the time series, such that we can observe what EOS trends we could observe by chance. For the time series that are taken from price ranges, we use a tipping point detection algorithm called *asdetect* (Boulton and Lenton, 2019) to determine where the tipping point is. This searches for anomalous changes in gradient in the time series of the system to determine where large changes are likely to have occurred, providing a detection value, the maximum of this being where the tipping is likely to occur.

We have included a Technical Appendix (Appendix A) which describes various aspects of our methods in greater detail.

**3 Results**

130 **3.1 The Full Marketplace**

We begin by exploring the proportion of advert views that are for EVs rather than non-EVs on Auto Trader's online platform (Fig. 1). This view share is low, particularly at the start in 2018 as the market is dominated by second-hand ICEVs. However, it is clear that there is an increase in the view share over the time period with 5 specific events that appear to drive an increase in interest in EVs for a short space of time (Fig. 1a). These events are detected by being at least 3 standard deviations away

135 from a trend calculated by the Kernal smoothing function (Fig. 1b red line, as described in the methods) and are as follows:

    i.    **4th February 2020** The UK Government announces a ban on sale of new petrol vehicles by 2035.

    ii.    **18th November 2020** The UK Government brings forward the ban on sale of new petrol vehicles to 2030.

    iii.    **29th September 2021** Potential HGV driver shortage, leading to uncertainty about petrol availability,

140         panic buying and fuel shortages in the UK.

    iv.    **10th March 2022** Spike in UK fuel prices associated with international fossil fuel volatility from Russian conflict in Ukraine.

    v.    **8th June 2022** Spike in UK fuel prices.

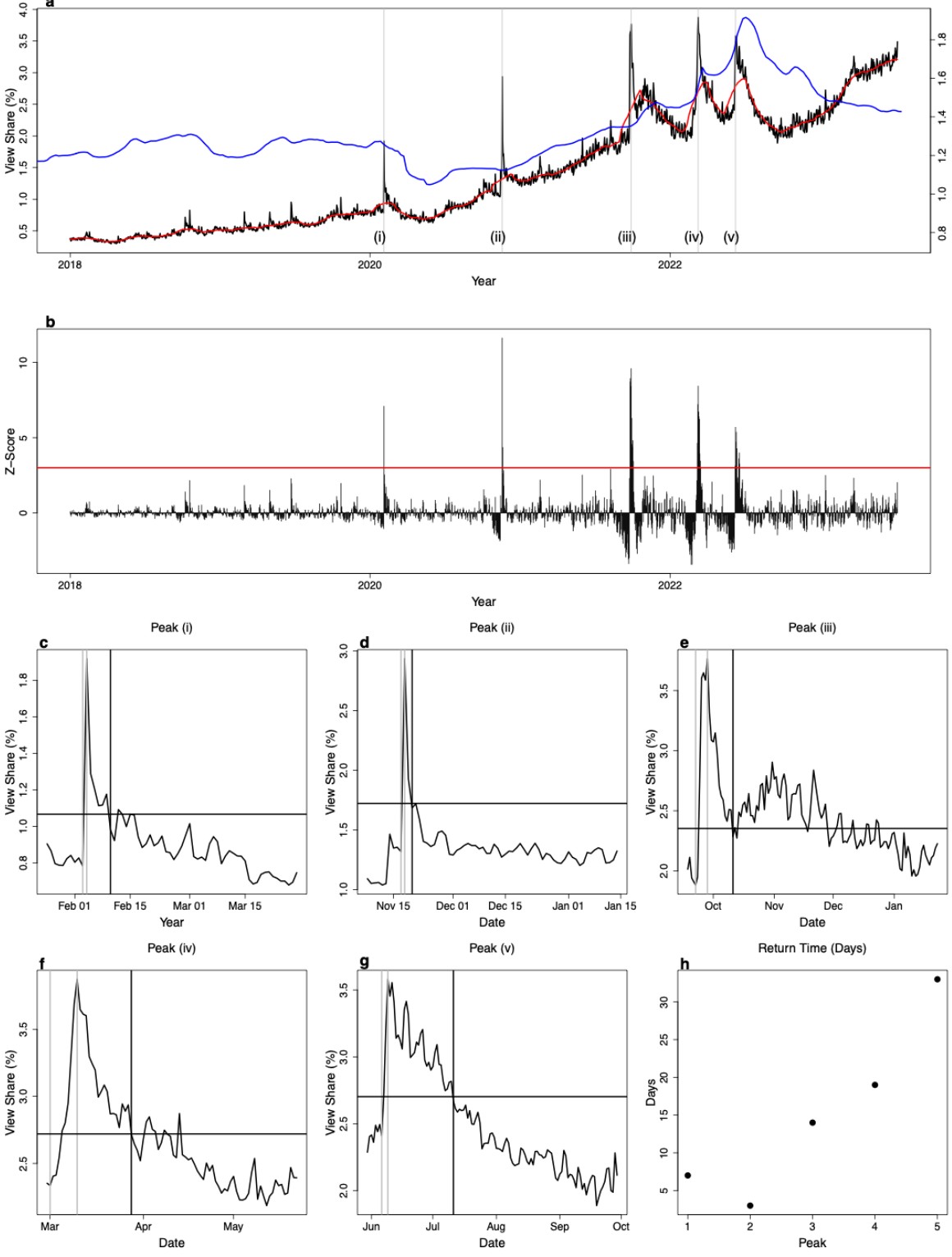

145

**Figure 1: Early warning signals on a daily time series of the percentage of views on Auto Trader's website that are for electric vehicles (rather than non-electric vehicles). (a) The time series of view share (black) with a Kernal smoothed time series as described in the Methods (red), alongside the weekly mean UK unleaded fuel price (blue). Marked in grey vertical lines (i-v) are specific external events detailed in the main text. (b) The original view share time series with the Kernal smoothed version subtracted (black). The red horizontal line at 3 standard deviations away from the mean (Z-Scores) shows the significance of the 5 peaks that are described in the main text. This detrended time series is used in the calculations in Fig. 2. (c)-(g) The return time from each event is calculated as the number of days it takes for the time series to decrease by 75% of the distance from the spike back to the pre-spike value. Dotted grey lines show the pre-spike and spike dates as vertical lines. The 75% value is shown as a horizontal black line, and the date this is reached by the vertical black line. (h) The number of days after the spike it took for the system to reach the 75% value for each spike.**

Plotting the weekly UK fuel prices (Fig. 1a; blue line) alongside the view share, makes clear the association with peaks (iv) and (v). Observing how long it takes for attention to return to normal after spikes (i)-(v) shows that the system is slowing down and the incumbent state of ICEV dominance of view share is losing stability over time. After the first two spikes (Fig. 1 c,d), for each successive spike (Fig. 1 e-g), there is a clear increase in return time (Fig. 1h). While we are unable to comment on the significance of a trend in return time over the 5 events, return time increases by more than a factor of 6 from event (ii) in November 2020, to event (v) in June 2022.

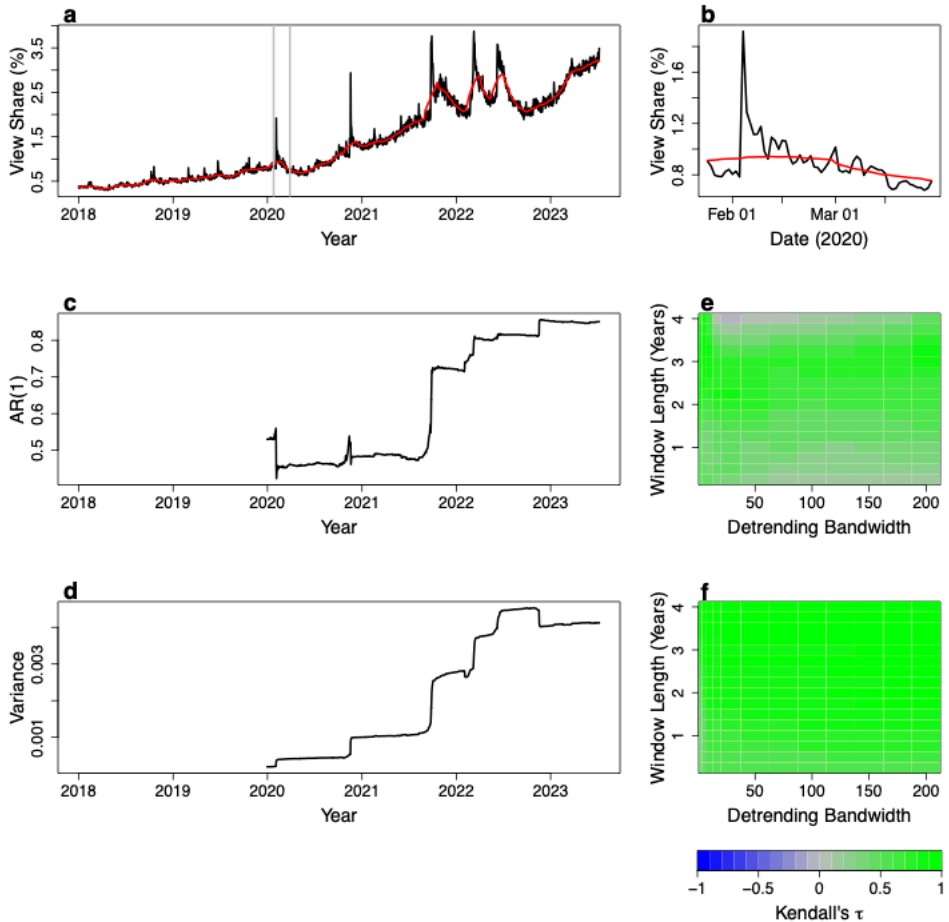

**Figure 2: Early opportunity signals on the (a) daily time series of the percentage of views on AutoTrader's website that are for electric vehicles. The Kernal smoothing function with bandwidth=50 (red, as shown in Fig. 1) is used to detrend this time series. However, it is difficult to detrend peaks fully; (b) shows the section in (a) between the two grey vertical lines. (c) AR(1) calculated from the time series in (a) once it has been detrended, using a moving window equal to 2 years (as described in the Methods) and plotted at the end of the window used to create it. (d) As in (c) but for variance. (e) and (f) show robustness tests for AR(1) and variance signals respectively, by varying the Kernal smoothing detrending bandwidth and window length used to calculate the signals. The Kendall's $\tau$ value for each combination is recorded in the contour plots, showing the tendency of the indicator time series.**

To look for temporal EOS on the EV view share time series, we first detrend as described in the Methods (Fig. 2a; red is the detrending line) to obtain a stationary series. After detrending the sharp spikes in attention remain (e.g., Fig. 2b). Calculating AR(1) on the residual time-series shows a significant increase over the whole time period (Fig. 2c; $\tau$=0.758, p=0.016, N=1286). The specific events generally cause upward jumps in the AR(1) (Fig. 2c). The sharp decrease in AR(1) at the start may be linked to the start of the COVID-19 pandemic. Variance also increases significantly (Fig. 2d; $\tau$=0.885, p<0.001, N=1286), and also appears to be influenced by the spikes in attention. As a check of robustness, we vary the detrending bandwidth for the

Kernal smoothing function and the window length used to calculate the EOS. For both AR(1) (Fig. 2e) and variance (Fig. 2f) we find robust increases across a range of window lengths and bandwidths. This includes using higher detrending bandwidths that better remove the spikes in attention. However, we are aware that these spikes in attention do mean that the noise driving the system is not randomly distributed white noise, which is a general requirement of these EOS. Nonetheless we argue they can still tell us something about the changing resilience of the system.

## 3.2 Looking for EOS in different niches of the market

During the time period covered by the dataset, cheaper EVs are expected to reach price parity with equivalent ICEVs sooner than more expensive ones. Hence, different niches of the market could tip at different times. The dataset presents us with the opportunity to look for EOS in different price niches. We begin by looking at the absolute number of adverts, as well as the absolute number of views (rather than view share). This allows us to see the evolution of EVs available without the influence of non-EVs. The analysis is split into different price bands (Fig. 3) in each year (noting that 2023 is year-to-date). For simplicity, we use £5,000 bands from £0-£5000, up to £125,000 and above. We have not accounted for inflation for simplicity and as we do not believe the proportion of car adverts that would end up in new brackets would be too small to affect the signals we observe over the period of time we are analysing, but could be a topic for future analysis over a longer time span.

It is clear from the number of adverts that more EVs become available over time and that these become particularly concentrated in the £20,000-£30,000 range. Also, the diversity of the market increases in terms of the spread of prices, with more expensive EVs becoming available in later years, which may signal a new niche emerging. In terms of advert views, whilst it is difficult to determine much information from 2018, in 2019 and 2020 there appears to be strong viewing activity in the £5,000-£10,000 and £35,000-£40,000 price band categories. Afterwards, this changes slightly with high views in the £20,000-£30,000 categories, alongside views in a £100,000+ category, this category likely to be an interest in a specific car each year. Up to and including 2022 the numbers of views are noisy across the categories, However, in 2023 we observe a change in the viewing habits, with a spike occurring in the £25,000-£30,000 with clearer drop-offs away from it. We have also looked at the advert share and view share per year for these price ranges and these results are discussed in the Supplement (Fig. S1).

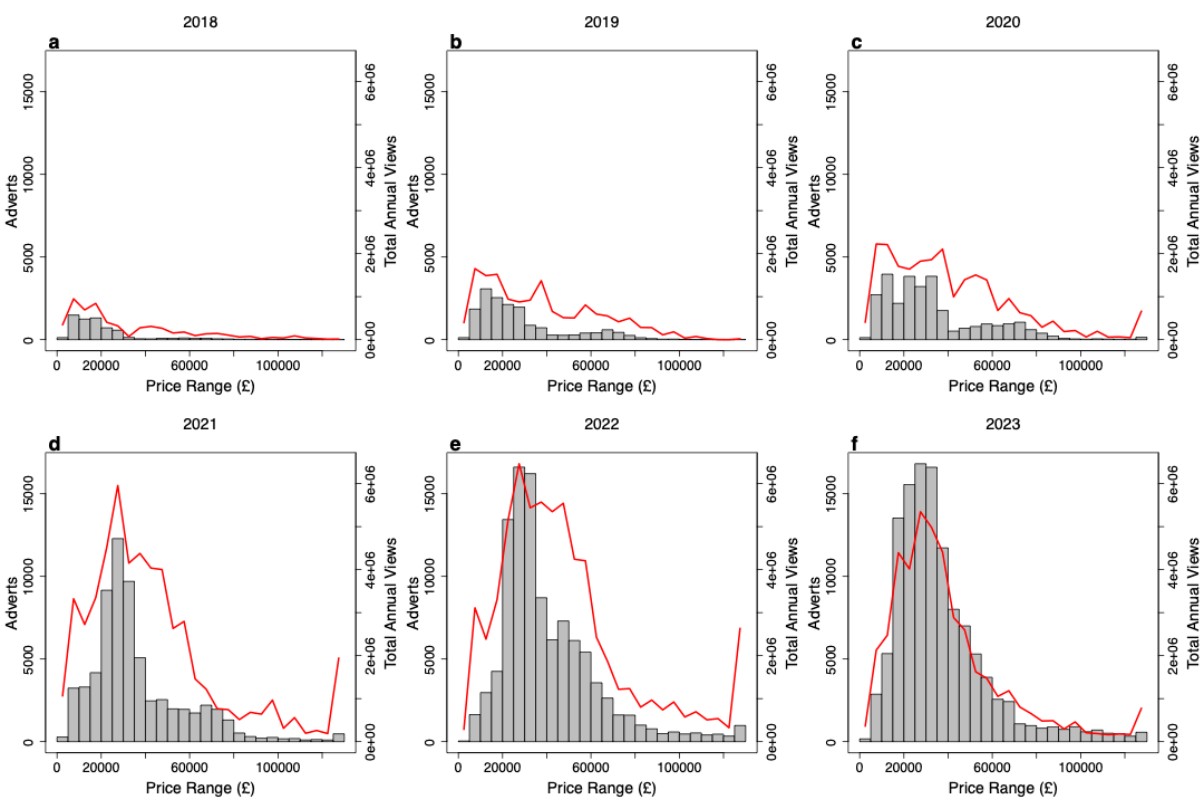

**Figure 3: Histograms showing the number of EV adverts per price range in (a) 2018, (b) 2019, (c) 2020, (d) 2021, (e) 2022 and (f) 2023 (to July), alongside the total EV advert views (solid red lines).**

Based on this apparent change in behaviour, we looked for EOS in these different price niches. Using the daily view share time series for adverts that are in each price category, we calculate AR(1) and variance on a 2 year window, using a detrending bandwidth of 50 (as in Fig. 1). We begin by looking at individual time series, those in the £10,000-£15,000 and £15,000-£20,000 categories. These appear to show a tipping point in behaviour from December 2022 (Fig. 4a,b). The time series of other price ranges are shown in Fig. S2. Increases in EOS are observed in both categories (£10,000-£15,000 AR(1) $\tau$: 0.536, variance $\tau$: 0.639, £15,000-£20,000 AR(1) $\tau$: 0.509, variance $\tau$: 0.733, particularly from mid-2021 (AR(1) $\tau$: 0.825, 0.756 from 1st June 2021, respectively). Also AR(1) reaches high values (approaching 1) which in the presence of large perturbations and a visible level of background noise would be expected to lead to tipping before the total loss of stability (at AR(1)=1). After assessing the time series in Fig. S2, we also calculate the view share on EV adverts under £35,000, finding that this too shows evidence of a tipping point (Fig. 4c) and similar EOS beforehand (AR(1) $\tau$: 0.509, 0.719 from 1st June 2021, variance $\tau$: 0.830, 0.698 from June 2021).

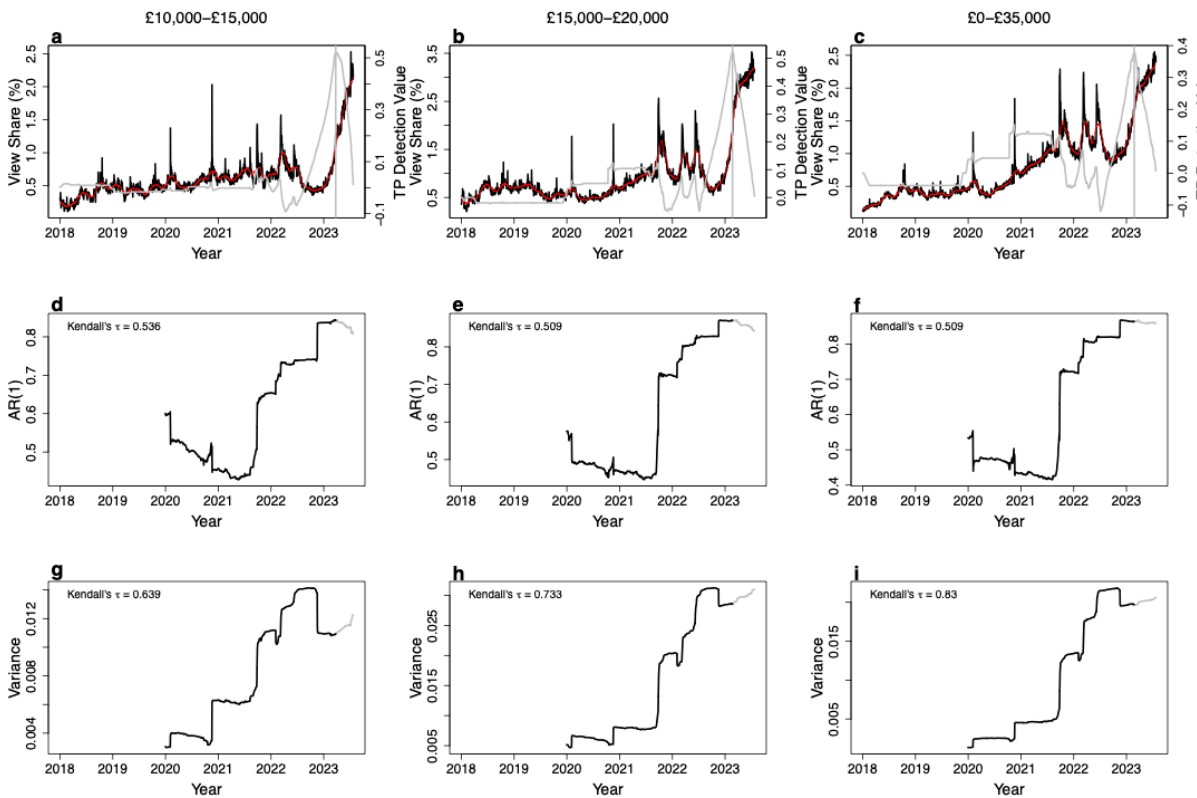

**Figure 4: Early opportunity signals carried out on EV view share for price bands (a) £10,000-£15,000 and (b) £15,000-£20,000, and (c) under £35,000 (black solid lines), chosen as time series that appear to show tipping points being crossed. Using the asdetect package to determine where the tipping point occurs (solid grey line), we cut the time series at the place we determine the view share time series begins to tip (vertical solid grey line). (d)-(f) AR(1) time series up to the vertical line for the time series in (a)-(c) respectively as a solid black lines, and afterwards as sold grey lines. (g)-(i) same as (d)-(f) but for variance.**

Looking across the full range of price bands, we calculate Kendall's $\tau$ correlation coefficient to measure the tendency of these two indicators for each time series. For certain higher price ranges, there are long periods of time where there are no adverts available for viewing and as such EOS are hard to calculate and prone to bias due to strings of zeros. Because of this, we use open circles to mark results from time series that have more than 10% of their length at 0% view share on Fig. 5.

Figure 5 shows that for AR(1), positive $\tau$ values are observed for advert views of EVs that cost up to around £70,000. Variance signals show similar behaviour, in price bands below £60,000. Together this suggests that the tipping point may be being approached in niches below this threshold. For AR(1), the strongest signal is observed at £55,000-£60,000 (0.763). For variance, the highest $\tau$ is in the £25,000-£30,000 category (0.886) where we noted an emergent behaviour in annual viewing numbers (Fig. 3). With this partitioned data, the signals generally lack significance (Fig. 5c,d), particularly for AR(1) (Fig. 5c).

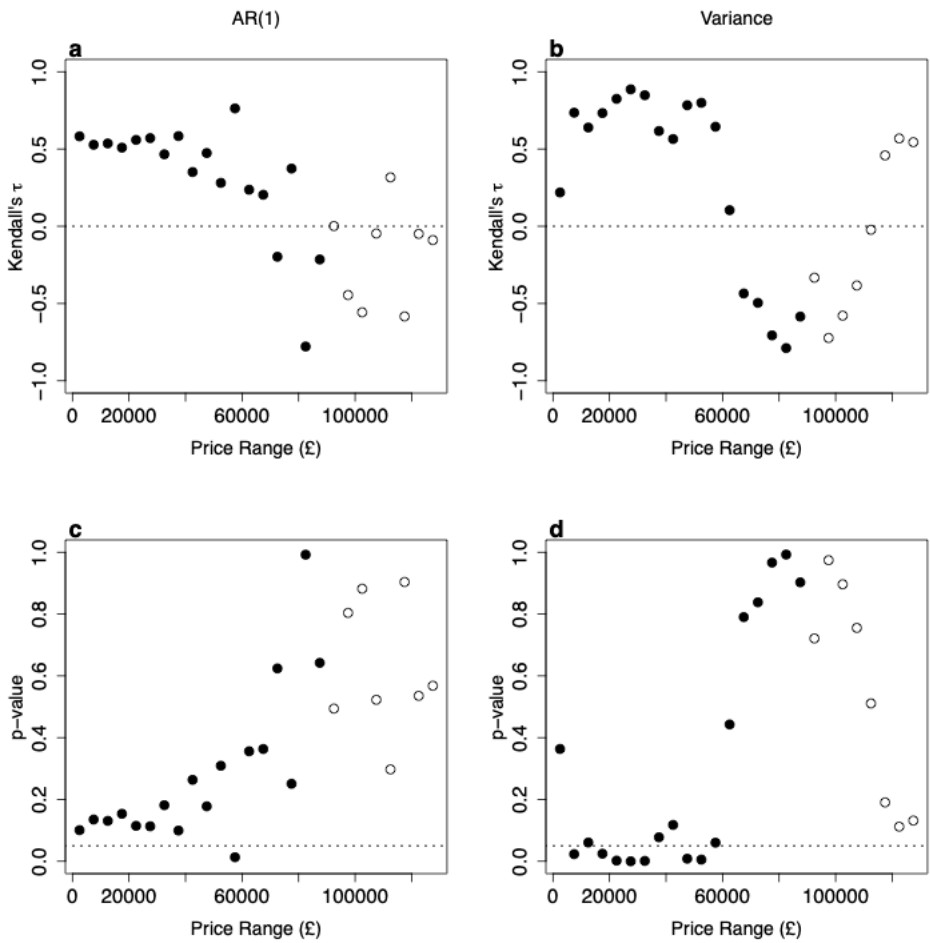

**Figure 5: Kendall's τ values to show the tendency of the EOS (a) AR(1) and (b) variance calculated on the view share time series per price range. These view share time series are shown in Fig. S2. Open circles denote price range time series that have no views for more than 10% of the time, due to there being no adverts available to view in those times. The dotted vertical line at τ=0 makes it clear where increases in trend are occurring. Significance of (c) AR(1) and (d) variance trends are shown as p-values as calculated with the use of null models, that destroy the memory in the time series. Any point below dotted vertical line at p=0.05 is deemed significant.**

## 4 Discussion

We find evidence that there are 'early opportunity signals' in EV advert views of the approach to a possible tipping point of greater attention on (mostly) second-hand EVs, and that in some price ranges such a tipping point has occurred. CSD is seen in both increasing return times from successive large perturbations and in long-term increases in AR(1) and variance. Events (i)-(v) cause pronounced spikes in attention to EV adverts, which are hard to detrend before calculating AR(1) and variance

and can influence these statistics. However, when successfully filtering out the spikes using a low bandwidth kernel smoother, significant increases in AR(1) and variance are still present.


The novel dataset we have used here focuses on public interest in EVs not the rarer purchase commitment reflected in sales data. We can expect advert view data to be more sensitive to specific events than sales data as it takes minimal effort to view adverts and it can be done multiple times, whilst buying a car is a rare, one-off event. Furthermore, advert views can react faster to events than sales, which always involve some delay. Searching for EOS in other datasets, such as market share, would

be beneficial. In other work, we have searched for EOS in EV sales data in major markets (China & Europe) with success (Mercure et al., 2024). Combining EOS indicators from multiple pertinent datasets could offer more of a 'dashboard' approach that can be used to assess confidence in opportunities to aid positive tipping.

We expect the reactions to government changes in policy to be different to the reaction to fuel shortages, for example. Fuel

events affect people in the present, whereas government policy affects people more than a decade into the future. As we tend to discount the future, one might expect a greater reaction to immediate events, all else being equal. Nonetheless, by looking at the *relative* changes in the system post-event, e.g. in our return time measure, we control for any such differences. Furthermore, we believe that the media may also play a role in the public's reaction to certain events; the last event for example occurs when fuel prices increase close to £2/litre which would have been considered a milestone in the media. The smaller

increase in fuel prices that occurs near the end of 2023 was unlikely to have the same media attention. This however, is something that requires further research.

The differences between social and natural systems regarding how they are forced can affect whether early opportunity (or warning) signals can be detected. Notably, social systems could be forced towards tipping extremely fast (e.g. by a globally

announced ban of ICEV sales) and this rapid change could eliminate the possibility of early no prior opportunity signals. In the case of the car market however, there have been several slow forcing factors such as emissions regulation on ICEVs being ramped up over time and different future ban times on new ICEVs being introduced at different times in different markets. These act to stimulate innovation, price declines and quality improvements in a globalised market.

Our results suggest a tipping point in UK public interest in (mostly) second-hand EVs is being approached and may have been crossed for some price ranges (Figure 4). A tipping point in sales as hypothesised in the introduction may be expected to follow. This is consistent with separate evidence analysed by Auto Trader (Auto Trader Uk, 2023). Unlike new vehicles, second-hand vehicle prices are dictated strongly by their age. An increase in supply of EVs in the secondary market occurred from March 2023, as cars bought on 3-year leases began to be available for resale, following an abrupt increase in new EV

sales that occurred back in 2020 (Mercure et al., 2024; Geels and Ayoub, 2023). Increased supply caused a pronounced decrease in second-hand EV price which has plausibly driven the upturn in view share, noticeable in Fig. 4. Comparing EVs

with their ICEV equivalent, Auto Trader find that price parity between 3-year-old vehicles in the secondary market has been reached in a number of niches early in 2023 (Auto Trader Uk, 2023).

**5 Conclusion**

We have provided evidence that early opportunity signals of positive tipping points of climate-friendly action are possible in a key social-technological system. Cars are responsible for around 13% of the UK's total greenhouse gas emissions. Further research should examine whether such early opportunity signals are more widespread in sectors of the economy with the potential for tipping points.

**Appendix A: Technical Appendix**

**Asdetect**

To search for potential tipping points or abrupt shifts in time series, we use the R package 'asdetect' which we created to search for anomalous gradients in time series (Boulton & Lenton, 2019). This splits the time series up into sections and fits linear regression models across each section. If the coefficient of the regression model falls above (below) 3 median absolute deviations away from the median coefficient, then a value of 1 if added (subtracted) to a detection time series. This step is
carried out for a range of window lengths used to create sections from 5, up to a third the length of the time series. The eventual detection time series is then divided by the number of window lengths used, creating a detection time series that shows the proportion of window lengths a shift was detected in.

**AR(1) and variance estimators**


AR(1) value estimation on a window of the data is carried out R using a base function, ar.ols(), which estimates autoregression coefficients with the use of ordinary least squares regression. Specifically, we use:

$$\text{ar.ols(x[i:(i+wl-1)], order.max=1, aic=FALSE)}$$


where x is the time series, i is the time step, wl is the window length, order.max=1 tells the function to only look for first order coefficients and aic=FALSE stops the best fitted model from being found (which would otherwise override the order.max=1 input).


This then estimates α in the equation:

$$x_{t+1} = \alpha x_t + \varepsilon_t$$

Where $\varepsilon_i$ are independently identically distributed white noise. The AR(1) coefficient is estimated using the estimator:

$$\hat{\alpha} = \frac{\sum_{i=1}^{n} x_{i-1} x_i}{\sum_{i=1}^{n} x_i^2}$$

Variance $\sigma^2$ is calculated using the R base function var() which uses the estimator:


$$\widehat{\sigma^2} = \frac{\sum_{i=1}^{n}(x_i - \bar{x})}{n - 1}$$

**Kernal smoothing function**

We use the R base function ksmooth() to detrend time series in our analysis, which creates a Kernal smoothing function, which is then subtracted from the original time series before early opportunities signals are calculated. This smoothed function can
be thought of as a weighted moving average calculation:

$$Y_i = f(x_i) + \varepsilon_i$$

Where $Y_i$ are the data and the $x_i$ are the dates. The function is estimated using the Kernal estimator:

$$\hat{f}_\lambda(x_i) = \frac{\frac{1}{n\lambda} \sum_{i=1}^{n} K\left(\frac{x - x_i}{\lambda}\right) Y_i}{\frac{1}{\lambda} \sum_{i=1}^{n} K\left(\frac{x - x_i}{\lambda}\right)}$$

With $\lambda$ the bandwidth which varies as described in the main text by defaults to 50. We use the box or uniform Kernal function K:

$$K = \frac{1}{2} \; for - 1 \leq x \leq 1$$

## Sensitivity and significance tests

Sensitivity tests are carried out by varying both the bandwidth $\lambda$ used in the Kernal smoothing function, above, and the window length used to calculate the early opportunity signals on.

Once a signal has been calculated with a bandwidth and window length combination, we use Kendall's $\tau$ rank correlation coefficient to determine the tendency of a signal, which calculates the proportion of calculated as:


$$\tau = \frac{2}{n(n+1)} \sum_{i<j} sgn(x_i - x_j) sgn(y_i - y_j)$$

Which measures the proportion of concordant pairs, with n being the total number of points in the time series. With x being time (which is always increasing) and y the signal time series.


To measure the significance of the $\tau$ value, we use null models which resample the time series once it has been detrended with the Kernal smoothing function, which keeps the overall mean and variance of the time series of residuals constant. Then, we calculate the early opportunity signals on these resampled time series and the resulting $\tau$ value. Repeating this 1000 times allows us to build a distribution of $\tau$ values that happen by chance, and from that we can calculate a p-value for the value we

observe in the original time series by measuring the proportion of null model values that are above the observed.

## Author Contributions

All authors developed the research and drafted the paper, CAB ran the analysis, all authors commented on the final text.

## Competing Interests

The authors declare that they have no conflict of interest.

## Financial Support

All authors were supported by the Bezos Earth Fund via the Global Tipping Points Report project.

## Acknowledgments

The authors acknowledge Auto Trader UK, particularly Louis Maxwell, for their generosity in sharing their datasets with them to allow this analysis to be carried out.

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
