# Peer review of "Early opportunity signals of a tipping point in the UK's second-hand electric vehicle market"

_EGUsphere, 2023_

## Referee Comment (RC2)

**ery-electric vehicles market share in European countries in Q1 2024**

ACEA (2024) Alternative Fuel Vehicle Registrations Data.

[Figure]

Market share ● BEV volume

BEV volume

90 K

50 K

Denmark · Sweden · Netherlands · Finland · Luxembourg · Belgium · France · Austria · Portugal · United Kingdom · Ireland · Germany · Romania · Hungary · Cyprus · Estonia · Latvia · Slovenia · Lithuania · Spain · Greece · Bulgaria · Poland · Italy · Slovakia · Croatia · EU27 · EU27+UK+NO

---

## Author Response (AR1)

**Review 1 – Lennart Baumgartner**
**General comments**
The authors apply statistical methods from the natural systems literature to a social system: the second-hand electric vehicle market in the UK. Based on statistical indicators, they identify "early opportunity signals" (EOS) of a "tipping point" in the overall EV resale market and different niche markets thereof.

They introduce a novel, interesting dataset of the second-hand EV market in the UK with high temporal resolution and details on different heterogeneous sub-markets (particularly different price ranges). They apply existing statistical methods to the new social context of this dataset that help identify non-linear dynamics in social systems. Clearly, these dynamics are very interesting to a number of research fields, as well as policymakers and the private sector. The work presents a meaningful contribution to the field, both in its methods and their application.

At the same time, the fact that the methods and concepts are novel in the field of social science means that they require precise definitions and detailed explanations. As discussed below, the authors should include more details to enable a comprehensive review of their method and make it both replicable and interpretable. I am also concerned about the statistical significance of some of the results.
Provided the authors provide these details and a second review of their method considers them adequate, the work is well worth publishing.

We thank the reviewer for taking their time to review the manuscript and for seeing it as a meaningful contribution to the field. Thank you also for the comments below which we have addressed in detail. Please find our responses below.

**Specific comments**
· The authors use a number of concepts without providing a clear definition thereof. First and foremost, it is not clear what constitutes a "tipping point". While this concept may be commonly applied to natural systems, it is unclear what it means in the case of social systems. Particularly in light of S-curve diffusion of technologies, whereby new technologies grow exponentially over several decades (as described by Grübler and cited by the authors), it is unclear how a tipping-point can be interpreted. Depending on the definition, this may have implications for the interpretation of the results and the methods used. Other terms that are used without clear definitions in the social context include "critical slowing down", or "early warning signals".

We have now added a clear definition of a tipping point, which in the case of S-curve diffusion of innovations is typically definable in terms of a critical mass of adoption, i.e. the point at which one more person adopting ultimately triggers the majority to adopt in a self-propelling fashion. Critical mass tipping points can arise from several distinct reinforcing feedback mechanisms, as reviewed in e.g. Lenton et al. (2022). For example, Everett Rogers in his classic text on diffusion of innovations describes the learning process whereby early adopters share knowledge with others in a population encouraging increased adoption (without any need for change in the qualities of the thing being adopted). This is pertinent to our case study. Alternatively, 'increasing returns' describes how adoption can increase the pay off for the next adopter, thanks to learning curves for a technology itself (e.g. involving learning-by-doing, economies-of-scale). This is being seen for EVs getting better and cheaper over time (both firsthand and secondhand) as well as the charging infrastructure and associated convenience of

use getting better with increased adoption. This information has been added to our manuscript in lines 32-39.

We have also added a definition of early warning/opportunity signals (statistical indicators that a tipping point is approaching) on line 58. We have also expanded more on the concept of critical slowing down (CSD), and as the system responds more sluggishly to perturbations, it is often referred to as losing 'resilience' as dampening feedbacks that maintain the incumbent state get weaker. We go on to say we are assessing CSD indicators of loss of resilience for the case of the incumbent state of a technology (ICEVs) dominating a market. The corresponding damping feedbacks are things that tend to maintain the status quo of ICEV market dominance, e.g. manufacturers and consumers resisting change. This expanded description can be found in lines 61-63.

· The authors make numerous statistical estimations based on their data. In order to make this work replicable to this and other data sets, it is essential to provide details of these methods. This is especially important in cases where the description of their method is somewhat ambiguous, such as the measurement of the "lag-1 autocorrelation (AR(1))" or the "detection algorithm called asdetect". I highly recommend the authors include a technical appendix to this work that shows the estimators used and formulas for their stochastic models. This appendix could include:

- o Data cleaning procedures (such as inflation adjustments)
- o The AR(1) and variance estimators and models
- o The kernel smoother used to de-trend the time-series data
- o The estimator for the Kendall tau correlation
- o Sensitivity analyses and significance tests

We didn't want the paper to be too technical and thus appeal to readers from a number of different fields so having this as a separate document is a good idea. A lot of the methods are using functions in R so we have expanded on what these are actually doing in the Technical Appendix (Appendix A). Although the sensitivity analyses and significance tests are described in the main text, they have been expanded upon in the Appendix.

It includes an expanded discussion on the asdetect package, the AR(1) and variance estimators and Kernal smoothing function used, and more information on the sensitivity and significance tests we carried out. It is referenced in the main text in line 127 at the end of the Methods section.

We do not adjust for inflation as we believe that the main reason for the behaviour we see is the influx of cars returned on 3 year leases. Adjusting for inflation across bands of £5,000 is not going to change the results but may cause some confusion when explaining. We have mentioned that we have decided not to do this in the discussion and added that it could be an avenue for future work (lines 192-194).

· The authors have made a substantial effort to test the statistical significance of their results. There are, however, a few open questions that should be addressed:

1. Figure 1 g): shows the return time for the 5 different spikes investigated. Given the limited data and the fact that there are no error bars

associated with each data point, it is unclear if the increase in return time is statistically significant.

We agree, and have amended the sentence at the end of the paragraph on line 161: 'While we are unable to comment on the significance on a trend in return time over 5 time points, return time increases…'

2. Figure 2: the authors use a variance and AR(1) estimation of the noise over time. Apart from the fact that it is not clear what they estimate in the AR(1) case, the variance estimation may be biased due to the fact that the noise could have heavy tails, as indicated by the spikes in the data. Also, note that most variance estimators assume i.i.d. normal residuals. This contradicts the earlier analysis of increasing return times.

We agree, and have put a caveat at the end of section 3.1 (lines 183-185) stating '...we are aware that these spikes in attention do mean that the noise driving the system is not randomly distributed white noise, which is a general requirement of these EOS. Nonetheless we argue they can still tell us something about the changing resilience of the system.'

3. Figure 2: The fact that the time-series data is bounded from below (by zero) may pose an issue to the noise estimation, particularly after de-trending the time-series. It is unclear how this affects the noise during the early phases of the data

The view share is bounded by 0, however it does not reach 0 with the minimum being 0.3%. As such we are confident that our results are unaffected by this bounding, even with detrending.

4. Figure 2: The use of a 2-year moving average window to estimate the AR(1) and variance means that individual points in the resulting time-series are strongly correlated. A single event, such as a spike in the data, will impact the variance estimation for all windows that include that event. This may explain the large jumps in the time-series observed in Figure 2 c) and d), as well as Figure 4. The significance test performed by the authors (re-shuffeling the time-series) may not be sufficient to proof statistical significance in the presence of strong correlation since it discards the autocorrelation structure of the data.

We understand the reviewer's concern, however by varying the window length used, smaller windows will have less events in, combined with the smaller bandwidth on the detrending function, we have done all we can to remove the effect of these events. Regarding the reshuffling, it is done on the residuals of the time series after detrending, and is done on the full time series rather than in windows and actually is designed to destroy the autocorrelation such that we can determine what autocorrelation trends could occur by chance. We have added 'These null models are designed to destroy the memory in the time series, such that we can observe what EOS trends we could observe by chance' on lines 121-122.

5. Figure 5: Given the estimation challenges described above, it is unclear how significant the results of Figure 5 are.

Hopefully with the above cleared up, the significance values in Fig. 5 are sufficient. We have added '...that destroy the memory of the time series.' in the figure caption when mentioning the null models now.

**Technical corrections**

· The caption of Figure 2 does not match what is shown.

Thank you for pointing this out, we have now altered the caption.

· In section 3.2, the authors mention that EVs have not yet reached price parity with ICEVs. It would be good to confirm that with newer Chinese EVs entering the European market and the fact that policy-enabled price parity has already been reached (see section 1 and section 4)

This may be a misunderstanding as we are talking about the second-hand vehicle market, and over the time frame of our dataset rather than presently. At the start of 3.2, we say that cheaper EVs are expected to reach price parity with equivalent ICEVs and have added 'During the time period covered by the dataset' at the start on line 187.

· Figures 2 and 4 are unclear on the cost units used (Figure 2 shows no unit on fuel costs, Figure 4 does not indicate any inflation adjustments)

We have added the fuel cost units on Figure 1. We have not done any inflation adjustments so have not changed Figure 4.

· The authors should comment on the nature of the second-hand vehicle market. The fact that second-hand vehicles are of different ages may distort their price.

In the discussion section talking about the return of 3 year lease EVs, we have added: 'Unlike new vehicles, second-hand vehicle prices are dictated strongly by their age.' on lines 282-283.

· The authors should comment on how they obtained the specific events listed in section 3.1 (and why they have excluded others)

The events in the time series cause spikes in attention that are at least 3 standard deviations away from the trend (when using the Kernal smoothing function). This is now shown as Figure 1b, and detailed in lines 134-135.

**Reviewer 2 – Gianluca Grimalda**

The paper aims to apply techniques of analysis from earth sciences - specifically, the identification of Early Warning Systems - to economic choices. The main idea is that the slowness with which a system comes back to the current steady state can be interpreted as a sign of "stress" that signals the future convergence to another steady state. This methodology applies, in particular, to systems that are characterized by tipping point dynamics. In this paper, the authors apply this technique to the UK second-hand electric car markets.

I think the methodology is fascinating and could provide valuable insights into social and economic dynamics. I think the paper could be improved if the authors tried to incorporate more the specificity of social and economic steady states into their analysis. I also think the authors should critically discuss the extent to which their analysis should really be indicative of a tipping point switch.

We thank the reviewer for their time reviewing our manuscript and for seeing its worth. Their constructive comments below have helped make the manuscript useful to a larger audience. Please find our responses to the comments below.

Major points:

1. The authors do not analyse in detail the theoretical underpinnings of what they call a switch from a market dominated by Internal Combustion Engine Vehicles (ICEV) to one dominated by Electric Vehicles (EV). I think the reader may benefit from a general treatment of a market equilibrium. A market equilibrium is characterized by equality of supply of demands with respect to a price level. From an economics perspective, the modelling of such a steady state would take into account that the purchase of a car is an investment, i.e., an action carried out today that will bring out benefits over the future. The buyer will take into account factors such as (a) the current price; (b) the future price level (if s/he wanted to resell the car); (c) the expected energy costs associated with the use of the vehicle; (d) accessibility of infrastructure, etc. The buyer may also consider non-economic factors such as (e) the environmental impact of using the car; (f) its popularity with the public, etc. The buyer will typically compare all the above factors with respect to several types of vehicles. At its simplest level, the buyer will compare the above factors for EV and CEV. The producer's decision will also involve a comparison between the profitability of investing into EVs or CEVs taking into account expected demand. One could think that this interaction is characterized not just by two equilibria but by an infinite collection of equilibria ranging from no use of EVs to universal use of EVs, all of which are characterised by equality in demand and supply at a certain price level. In order for a product to "dominate" a market, one would want to introduce some non-linearity in the system. I think the most obvious way would be to assume a model of technological change characterised by a logistic-like curve, in which technological progress is steep when the technology is "young", and it reaches a plateau when it is "mature" (see references below). The mature technology can then be overtaken by a new young technology. In the case of ICEV, the switch from increasing returns to scale to decreasing returns to scale in technological progress may be determined by institutional or economic constraints in access to a certain natural resource - fossil fuels in this case. This model would create a situation of multiplicity of equilibria with a tipping point dynamics. In general terms, we could also model this situation as a "coordination game" with a representative consumer and a representative producer, in which the two equilibria are characterised by high consumer demand for EV and high producer investment in EV, or low consumer demand for EV and low producer investment in EV. A coordination game is characterised by a tipping point dynamic.

These points are very well taken and tally with our own previous writing e.g. in Lenton et al. (2022). We have expanded the introduction to include information on S-curve diffusion of innovation, with a host of reinforcing feedbacks supporting the self-propelling transition, including social contagion, learning-by-doing, economies-of-scale, and a coordination game. EVs (and their Li-ion batteries) are now displaying all of these reinforcing feedbacks (several of which produce strong increasing returns to adoption), whilst several 'lock-ins' to ICEVs and their manufacture are also apparent. There is the clear coordination game of switching to a different powering/fuelling infrastructure alongside changing the engine technology. As production is scaled up, EVs can become cheaper to manufacture (and purchase) than ICEVs and are already considerably

cheaper to run, with better performance. Hence all the conditions for alternative stable states and a tipping point are met. This can be found in lines 32-50.

The UK secondhand market is particularly interesting as purchase price parity (for equivalent EVs and ICEVs) has recently been achieved, ahead of price parity for new vehicles as described in lines 286-288.

2. The analysis provided by the author is fascinating and suggestive of possible tipping point dynamics. However, it would take a big leap of faith to believe that the system will now switch to an EV-dominated market on the basis of four episodes of critical slowing down in the advert market. In the lack of extensive analyses of similar social dynamics, it becomes impossible to ascertain whether this evidence is sufficient or not. Relatedly, I felt that the claim that "a tipping point has been crossed for some price ranges" (line 221) is a bit of an overstatement. I suppose the authors could slightly reposition their paper in saying that they provide pioneering analyses of early 'opportunity' signals (EOS) predicting tipping points in social systems, which will have to be complemented by re-examination and further research to be thoroughly evaluated.

We have hopefully built a stronger case that a tipping point can exist (see above).

We agree that a tipping point is not yet firmly established and have rephrased accordingly (e.g. toning down the statements to say the tipping point 'may have' been reached in what is now line 280 from 221). There is separate information (from AutoTrader) that purchase price parity of equivalent EV and ICEV models has been reached in some price ranges towards the end of the timeseries we analyse. We recognise that other factors, as noted by the referee, can influence a 'critical mass' tipping point of adoption, but nevertheless purchase price is an important one.

3. It could be that the authors are right and product adverts are indeed successful EOS predicting tipping points in socio-economic systems. However, I am left wondering whether other signals may be used in addition to product adverts. In the light of the above market analysis, one could think of supply-side indicators, such as stock exchange fluctuations, firms' investment, firms' sales. From the demand side, google searches may also be used. All these indicators may converge into a dashboard of EOS. To the very least, the authors should critically discuss why they opted for the specific indicator they used, as well as the pros and cons of using a dashboard of indicators rather than just one indicator.

These are good suggestions that we have also been considering. In a separate submitted paper we have analysed (firsthand) market share data for EVs and ICEVs for a number of countries and also found early opportunity signals in major markets including China and European ones (Mercure et al. 2024), which we now mention in line 261-262. We have also added 'Combining EOS indicators from multiple pertinent datasets could offer more of a 'dashboard' approach that can be used to assess confidence in opportunities to aid positive tipping.' to the discussion on lines 262-263.

4. It may also be helpful to characterize other differences between social systems and physics systems. The authors state that "EWS generally assume a timescale separation; a long term, slow forcing towards tipping" (line 44). As the authors already hint at, in social systems the forcing could be instantaneous and still trigger a switch. Suppose that all world governments announced the ban of fossil fuels from tomorrow. We can be sure that the system would tip overnight. With social systems, the forcing may be given by expectations over future states, thus possibly changing

instantaneously. This characterisation probably does not apply to the present case study, but it may be worth bearing it in mind.

We have now added some characterisation of pertinent differences between physical and social systems. We recognise the possibilities and also that very rapid forcing past a tipping point, if there were no steady changes towards the tipping point beforehand, could produce cases of tipping without early warming/opportunity signals. However, we note that there are several slower forcing factors in our chosen case. For example, emissions regulation on ICEVs have been progressively ramped up over time (e.g. in the EU) and bans on petrol/diesel car sales are not implemented simultaneously everywhere, rather they are occurring in different countries at different times, but all act to stimulate innovation, price declines and quality improvements in a globalised market (to varying degrees, depending on country market size). This discussion can be found in lines 273-278.

Minor points:

1.  In the conclusions, I notice a more modest definition of what the authors mean by tipping point, which is inconsistent with what given in the introduction: "a tipping point in UK public interest in (mostly) second-hand EVs is being approached".

To make the distinction between these two tipping points, we have changed the sentence in the discussion to 'A tipping point in sales *as hypothesised in the introduction* may be expected to follow.' (line 281).

2.  Maybe an "objective" criterion should be given to define what a price spike is? This would be good also for replication/extension of the methodology.
3.  Relatedly, maybe some other petrol price spikes did not lead to any appreciable change in advert views? Is there a way we can rule out the choice of spikes was somehow "biased"?

We are answering these two comments together. Based on the other reviewer's comments, we have described more clearly how we chose the spikes in attention in the view share time series, based on change that is 3 standard deviations away from the trend of the time series. This is shown in Figure 1b and detailed in lines 134-135.

Spikes in fuel prices are used more as a descriptive entity to determine why we see spikes in attention, and we agree that there are some increases in fuel prices that do not appear to directly link to spikes in attention. We can only speculate that the media in the UK also drive this and that particularly for the last spike (v), approaching the £2/litre mark was a milestone that would have grabbed attention. The increase afterwards towards the end of 2023 occurring in a decreasing price trend, for example, was unlikely to have caused the same attention. This is described in the discussion in lines 268-271.

4.  In the analyses reported at lines 136 and ff, what is N?

We have now added N=1286, assuming the reviewer is referring to the length of the time series of AR(1) and variance.

5.  I think it would be good to show the evolution of EV cars in the UK market, for the reader to appreciate actual market evolution.

We would have to pay for the data for this. However, we have now included some information on how both the EV sales market has evolved in the UK, as well as enquiries about cars from the adverts in our dataset have evolved over the time period of the dataset: 'The EV market in the UK has grown over the last few years, with new EV sales making up 0.7% of all vehicle sales in 2019, up to 16.5% in 2023. Specific to our analysis, interest in EVs is also growing; 0.9% of all advert enquiries on AutoTrader were for EVs in 2019, however this had increased up to 7% by 2023, the share doubling from 3.5% in 2022.' has been included on lines 82-85.

6.    Line 11: ICEV not previously defined.

Thank you for spotting this, this has now been included in the abstract.

References:
Dosi, G., Moneta, A., & Stepanova, E. (2019). Dynamic increasing returns and innovation diffusion: bringing Polya Urn processes to the empirical data. *Industry and Innovation*, *26*(4), 461-478.
Kucharavy, D., & De Guio, R. (2011). Application of S-shaped curves. *Procedia Engineering*, *9*, 559-572.
Silverberg, G., & Verspagen, B. (1994). Collective learning, innovation and growth in a boundedly rational, evolutionary world. *Journal of Evolutionary Economics*, *4*, 207-226.

We have now cited these in our introduction in line 43.

References cited in our responses:

Lenton, T. M., Benson, S., Smith, T., Ewer, T., Lanel, V., Petykowski, E., ... & Sharpe, S. (2022). Operationalising positive tipping points towards global sustainability. *Global Sustainability*, *5*, e1

Mercure, J.-F., Lam, A., Buxton, J., Boulton, C., and Lenton, T. (2024) Evidence of a cascading positive tipping point towards electric vehicles. *Nature (in review)* 10.21203/rs.3.rs-3979270/v1,

---

## Author Response (AR2)

Editor comments

The reviewer is happy with the edits that have been made. I would like to offer the authors the chance to edit the manuscript now (also given the delay in review) to update with any new references or figures as suggested by the reviewer.

We thank the editor for giving us the opportunity to respond to the comments below. We have updated with new reference to IEA (2024) and associated data.

Reviewer comments

I am satisfied with the revisions provided by the authors, although I find the queries posed by the other reviewer as valid.

We thank the reviewer for rereviewing our manuscript and for their suggestions below.

A minor point is that including the analyses carried out in the other paper submitted to Nature would have obviously made this paper stronger. But of course a publication in Nature obliges. If the paper does happen to be published in Nature, though, the originality of the present paper is obviously reduced. I would urge the authors to mention the findings from the other paper. The best place would probably be in the discussion. Something like "Results from the present paper are strengthened by findings in a companion paper...".

The preprint of the other paper is already cited and the content mentioned in line 260. We have strengthen this link by stating that it is a companion paper and providing more detail: "In other work in a companion paper, we have searched for EOS in EV sales data in major markets (China & Europe), by successfully searching for critical slowing down signals in ICEV market share time series (Mercure et al., 2024)." This other paper is about to be resubmitted and as such the reference for it may change depending on the timeline of the publication of the final version of this paper – but the reference to the preprint will remain accurate and accessible.

Another point the authors may wish to discuss is the graph from Our World in Data showing the diffusion of EV in many markets around the world: https://ourworldindata.org/electric-car-sales . The graph for the UK shows a clear plateau at a relatively low level of EV market share. This made me somehow question whether we have actually reached a tipping point in the UK. The same is true for the EU, while in China the evolution of EV market share seems compatible with convergence to a steady state dominated by EVs. I leave it to the authors whether they want to mention this graph in their discussion or not and discuss how they relate to their data.

2023 proved an unusual year in several key car markets, including the UK, where the whole market grew rapidly in a delayed recovery from the pandemic. While EV sales continued to grow, the rapid recovery of ICEV sales has caused what looks like a stalling in the market share of EVs. However, having recovered, the overall market and ICEV sales are unlikely to keep growing in these developed economies. Thus, one should not

over-interpret the temporary stalling of EV market share as signalling a more persistent stall. It is more informative to look at the actual sales of EVs, which as the OWID source shows have continued to grow markedly through 2023.

To the end of the discussion, we have added: "There appears to have been a plateau in the market share of new EVs in the UK in 2023 (IEA, 2024). However, when viewing the raw number of EVs that are currently in use in the country, there was a marked increase from 950,000 (3% of the total fleet) in 2022, to 1.58 million (5% of the fleet) in 2023. The total new car market exhibited unusual ~17% growth in 2023, in a delayed recovery from the turbulence of the COVID pandemic and supply chain issues. This is unlikely to persist. Indeed, in 2024 thus far the total market has only grown ~2% whereas EV sales have grown 14% and their market share has increased to over 18% (Auto Trader, UK, personal communication, 28[th] November 2024)."